# LATENT REASONING VIA SENTENCE EMBEDDING PREDICTION

## ABSTRACT

Autoregressive Language Models (LMs) generate one token at a time, yet human reasoning operates over higher-level abstractions—sentences, propositions, and concepts. This contrast raises a central question: can LMs likewise learn to reason over structured semantic units rather than raw token sequences? In this work, we investigate whether pretrained LMs can be lifted into such abstract reasoning spaces building on their learned representations. We present a framework that *adapts* a pretrained token-level LM to operate in *sentence space*, by autoregressively predicting continuous embeddings of next sentences. We explore two embedding paradigms inspired by classical representation learning: **semantic** embeddings, learned via autoencoding to preserve surface meaning; and (ii) **contextual** embeddings, trained via next-sentence prediction to encode anticipatory structure. We evaluate both under two inference regimes: DISCRETIZED, which decodes each predicted embedding into text before re-encoding; and CONTINUOUS, which reasons entirely in embedding space for improved efficiency. Across four domains—mathematics, logic, commonsense, and planning—contextual embeddings under continuous inference show competitive performance with Chain-of-Thought (CoT) while reducing inference-time FLOPs in average by half. We also present early signs of scalability and modular adaptation. Finally, to visualize latent trajectories, we introduce *SentenceLens*, a diagnostic tool that decodes intermediate model states into interpretable sentences. Together, our results indicate that pretrained LMs can effectively transition to abstract, structured reasoning within latent embedding spaces.[1]

## 1 INTRODUCTION

Autoregressive Language Models (LMs) have achieved remarkable success on complex reasoning tasks through a simple objective: Next-Token Prediction (Bengio et al., 2003). This success is further amplified by Chain-of-Thought (CoT), which generates explicit intermediate reasoning steps to guide the model (Wei et al., 2022). Recent advancements demonstrate substantial gains in performance by scaling inference-time computation even further (Jaech et al., 2024; Guo et al., 2025). However, next-token prediction requires generating long reasoning chains one token at a time, making it computationally inefficient. Also, it remains unanswered whether reasoning at such granularity is genuinely optimal.

While token-level generation has driven recent progress, human cognition typically operates over higher-level abstractions—such as concepts, propositions, or full sentences (Fodor, 1975; Mercier & Sperber, 2011; Bengio, 2019). Prior works suggest that language models may similarly benefit from operating at these higher levels, potentially enabling more structured and computationally efficient reasoning (team et al., 2024; Tack et al., 2025).

In this paper, we investigate whether pretrained language models can effectively build higher-level representations directly by abstracting over their existing token-level representations, *without* the prohibitive cost of pre-training from scratch. Specifically, we introduce a framework that repurposes pretrained next-token Transformers to reason in a latent sentence-level embedding space. Instead of producing outputs token-by-token, our approach predicts continuous embeddings for entire sen-

---

[1]Our code is available here.

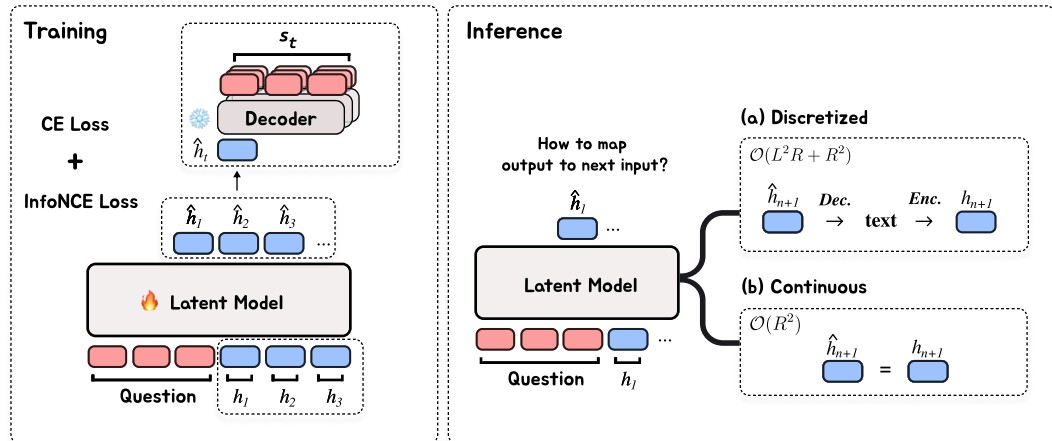

Figure 1: Sentence-level reasoning framework. **Training**: the latent model reads the question tokens and previous embeddings, predicts $\hat{h}_t$, and a frozen decoder reconstructs $s_t$; **Inference**: embedding can be rolled forward by **(a) Discretized**: decode → text → encode or **(b) Continuous**: pass-through.

tences, which can be decoded back into natural language yet primarily function as abstract conceptual representations.

To systematically explore viable latent representations, we draw inspiration from the well-established dichotomy in classical representation learning between reconstruction-based and prediction-based methods (Dai & Le, 2015; Kiros et al., 2015; van den Oord et al., 2019). We define two embedding paradigms: (1) **Semantic** embeddings, which prioritize preserving textual fidelity through autoencoding, and (2) **Contextual** embeddings, which focus on capturing predictive context via next-sentence prediction.

We evaluate models trained with these embeddings under two inference regimes: DISCRETIZED, which decodes each predicted embedding into natural language before re-encoding it as the next input, and CONTINUOUS, which performs reasoning entirely within the continuous embedding space. Our empirical findings demonstrate that contextual embeddings consistently outperform semantic embeddings across diverse reasoning domains including mathematics, logic, commonsense, and planning tasks. Notably, contextual embeddings using Continuous inference show competitive performance to token level Chain of Thought reasoning while reducing inference time computational cost by half in average.

Finally, we introduce **SentenceLens**, a diagnostic tool that translates intermediate hidden states into readable sentences, thus providing intuitive transparency into the model's internal reasoning trajectories. Overall, our analysis provides initial evidence that pretrained inductive biases acquired from token level modeling can be effectively adapted to structured, abstraction level reasoning within latent embedding spaces.

## 2 SENTENCE EMBEDDINGS FOR AUTOREGRESSIVE MODELING

Unsupervised and semi-supervised sequence representation learning has predominantly evolved along two primary paradigms: *reconstruction-based* and *prediction-based* methods (Dai & Le, 2015; Kiros et al., 2015; van den Oord et al., 2019). Both methodologies have demonstrated strong empirical performance, yet each emphasizes distinct representational strengths. Reconstruction-based methods, typically employing autoencoder architectures, excel at semantic fidelity by explicitly encoding and reconstructing input sequences (Dai & Le, 2015), whereas prediction-based methods prioritize capturing contextual semantics by modeling relations to subsequent sequences (Kiros et al., 2015).

Previous research suggests that the optimal embedding strategy varies significantly depending on the target application (Hill et al., 2016). In this light, we systematically explore both embedding

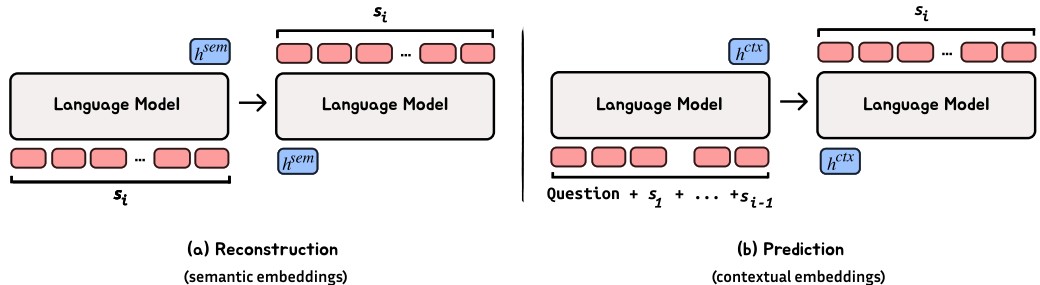

Figure 2: Illustration of the different types of sentence embeddings used in our framework.

paradigms within the context of sentence-level autoregressive modeling. Specifically, we adapt an autoregressive Language Model autoencoder framework to construct and evaluate two distinct embedding approaches: **semantic embedding**, derived through reconstruction objective, and **contextual embedding**, derived through predictive objective.

## 2.1 SENTENCE EMBEDDING CONSTRUCTION

To ensure scalability and avoid vocabulary constraints inherent to discrete codebooks van den Oord et al. (2018), we utilize a continuous embedding space. This approach facilitates flexible representational capacity scaling with embedding dimensionality (Kuratov et al., 2025). We build upon the autoencoding framework proposed by ICAE (Ge et al., 2024) and adapt a decoder-only Transformer (e.g., GPT-2), employing shared parameters for encoding and decoding: $\theta_{\text{ENC}} = \theta_{\text{DEC}}$.

Given an input sequence $x = (x_1, \ldots, x_N)$, the encoder produces a sequence of hidden states $H = (h_1, \ldots, h_N)$. We then define the embedding $h^{[-1]} := h_N$ as the latent representation of the entire input sequence. This embedding conditions the decoder, trained autoregressively with cross-entropy loss:

$$\hat{y} = \theta_{\text{DEC}}(h^{[-1]}) \quad \text{and} \quad \mathcal{L}_{\text{CE}} = -\sum_{t=1}^{N} \log p(y_t \mid y_{<t}, h^{[-1]})$$

Note that most reasoning tasks consist of a question or instruction $q$, followed by an ordered sequence of reasoning steps $(s_1, \ldots, s_n)$. In this light, we construct training examples tailored to each embedding type as follows (See Figure 2):

**Semantic embeddings.** Each reasoning step $s_i$ independently forms the input and reconstruction target $x = y = s_i$. Training this way ensures the embedding $h^{[-1]}$ encapsulates complete and detailed semantics of the individual reasoning step.

**Contextual embeddings.** We form context–target pairs, where context $x$ includes the question and preceding reasoning steps $(q, s_1, \ldots, s_{i-1})$, and the target is the current step $y = s_i$. Thus, embeddings must capture predictive cues essential for reasoning step generation.

Optionally, to bridge semantic fidelity with predictive abstraction, we also try a contrastive regularization loss (InfoNCE), aligning contextual embeddings closer to corresponding semantic embeddings:

$$\mathcal{L}_{\text{InfoNCE}} = -\log \frac{\exp\left(\text{sim}(\hat{z}_i, z_i^{\text{sem}})/\tau\right)}{\sum_j \exp\left(\text{sim}(\hat{z}_i, z_j^{\text{sem}})/\tau\right)},$$

where $\hat{z}_i$ is a contextual embedding and $z_i^{\text{sem}}$ a semantic embedding. Negative examples $z_j^{\text{sem}}$ are sampled within the batch. We refer to this regularized approach as **Contextual-Contrastive (CTX-C)** and the unregularized baseline as **Contextual-Base (CTX-B)**.

| | RECONSTRUCTION | PREDICTION | | |
|---|---|---|---|---|
| DATASET | SEMANTIC (EM) | CTX-B | CTX-C | COT |
| GSM8K | 98.5 | 42.0 | 42.1 | 43.4 |
| CSQA | 98.5 | 33.8 | 35.1 | 35.7 |
| PROSQA | 100.0 | 80.2 | 75.3 | 77.5 |
| BLOCKSWORLD | 100.0 | 89.9 | 90.1 | 84.3 |

Table 1: Performance of Semantic and Contextual Embeddings across datasets. For Semantic embeddings, we report exact match (EM). For Contextual embeddings, we compare final-answer accuracy (ACC) under different decoding schemes: CTX-B (unregularized), CTX-C (contrastive), and CoT (language-level chain-of-thought).

## 2.2 EMBEDDING EVALUATION

**Setting** We evaluate our framework using GPT-2 across four distinct reasoning domains: mathematical reasoning (GSM8K (Cobbe et al., 2021)), commonsense reasoning (CommonsenseQA (Talmor et al., 2019)), logical reasoning (ProsQA (Hao et al., 2024)), and planning (Blocksworld). For each domain, we train on the respective training split and report accuracy on the corresponding test set, analyzing how well our framework generalizes across diverse linguistic subspaces. (*i.e.,* mathematical expressions, natural language, etc.)[2] See Appendix C and G for more details.

To evaluate *semantic* embedding's performance, we compute exact match (EM) between the original reasoning step $s_i$ and the decoder output, assessing how faithfully the model reconstructs unseen steps. For *contextual* evaluation, as there could be multiple correct *next* steps that could lead to the correct answer, we roll out the model autoregressively: at each step, the generated output $y$ is appended to the current input $x$, continuing until a terminal answer is produced. The final answer is then compared against the ground-truth answer. Results are reported in Table 1.

**Results** Across all domains, we observe that the autoencoder successfully restores the original sentences with high fidelity. This aligns with findings from Kuratov et al. (2025), who show—both theoretically and empirically—that language models can compress a substantial number of tokens into compact representations. Yet, as we form CommonsenseQA (CSQA) task's SEMANTIC embedding using a subset of Fineweb-Edu corpus (~100k documents), we highlight that larger language space (compared to synthetic, constrained, i.e. ProsQA and Blocksworld) involves a higher difficulty.

In the *Contextual* configuration, model performance approaches that of the CoT baseline on three out of four benchmarks, and notably surpasses it on BLOCKSWORLD across both contextual variants. Introducing the contrastive alignment term (CTX-C) leads to a nuanced pattern: scores remain largely unchanged on GSM8K and BLOCKSWORLD, improve modestly on CommonsenseQA, but decline on ProsQA. These trends appear closely tied to each task's underlying semantic structure.

CommonsenseQA questions exhibit substantial lexical variety, so anchoring each latent vector to its semantic counterpart helps tame surface variability. In contrast, ProsQA benefits from simultaneously tracking multiple evolving states; consequently, enforcing a single semantic target at each step restricts its representational flexibility, which is consistent with earlier findings (Hao et al., 2024; Deng et al., 2024). GSM8K and BLOCKSWORLD are highly symbolic and lexically sparse—thus, the baseline contextual embedding already forms an unambiguous mapping, leaving little space for improvement through additional regularization.

---

[2]For CSQA *restoration*, we trained on a small subset of FineWeb-Edu (Penedo et al., 2024) due to small CSQA training set.

## 3  SENTENCE-LEVEL REASONING MODEL

Given the strong reconstruction and predictive capabilities of semantic and contextual embeddings, we now present a framework that leverages these embeddings for sentence-level reasoning. (Figure 1)

### 3.1  ARCHITECTURE

We adapt a pretrained decoder-only Transformer (Vaswani et al., 2023) to operate directly over continuous sentence embeddings instead of discrete natural language tokens. We refer to this model as the *Latent Model* $\theta_{\text{LAT}}$. Formally, given a natural language question $q$ and a sequence of latent embeddings corresponding to previously generated sentences $h_1, \ldots, h_t$, the latent model predicts the embedding for the next sentence:

$$\hat{h}_{t+1} = \theta_{\text{LAT}}(q, h_{\leq t}).$$

At inference time, predicted embeddings $\hat{h}_{t+1}$ are mapped to the next input embedding $h_{t+1}$ using a mapping function $\mathcal{M} : \mathbb{R}^d \to \mathbb{R}^d$, where $d$ denotes the embedding dimensionality:

$$h_{t+1} = \mathcal{M}(\hat{h}_{t+1}).$$

This process continues autoregressively, forming a latent embedding trajectory that encodes the progression of reasoning steps. At each step, a sentence decoder $\theta_{\text{DEC}} : \mathbb{R}^d \to \mathcal{T}$ can decode latent embeddings back into natural language text. However, decoding intermediate reasoning steps is optional; embeddings can remain in their latent form to enhance computational efficiency, particularly when only the final answer is required. To this end, a lightweight termination classifier can evaluate each predicted embedding $\hat{h}_t$ to determine when reasoning should conclude.[3]

### 3.2  TRAINING

A natural approach for this task is to train the transformer model to generate sentence embeddings by minimizing the Mean Squared Error (MSE) between predicted and target embeddings. However, a single context often allows for several valid yet distinctly different continuations. (team et al., 2024). Under these conditions, MSE tends to blend these varied possibilities into a single averaged representation, thus blurring meaningful variation.

To address this, we employ a cross-entropy (CE) loss calculated over natural language targets generated by a frozen decoder. This encourages predicted embeddings to align with the manifold defined by such decoder:

$$\mathcal{L}_{\text{CE}} = -\sum_{t=1}^{n-1} \log p\big(s_{t+1} \mid \theta_{\text{DEC}}(\hat{h}_{t+1})\big).$$

During training, the latent model conditions on the question $q$ and ground-truth sentence embeddings $h_i$, each computed using a fixed encoder $\theta_{\text{ENC}}$. Additionally, to enhance the alignment between predicted and teacher-forced embeddings, we incorporate an InfoNCE loss (van den Oord et al., 2018):

$$\mathcal{L}_{\text{InfoNCE}} = -\sum_{t=1}^{n-1} \log \frac{\exp\big(\text{sim}(\hat{h}_{t+1}, h_{t+1})/\tau\big)}{\sum_j \exp\big(\text{sim}(\hat{h}_{t+1}, h_j)/\tau\big)}.$$

The overall training objective combines both terms: $\mathcal{L}_{\text{overall}} = \mathcal{L}_{\text{CE}} + \lambda\, \mathcal{L}_{\text{InfoNCE}}$. To further improve training stability, we include shallow projection layers between the encoder output and latent model input, and between the latent model output and decoder input.

---

[3] We use an oracle termination classifier for simplicity. See Appendix F for more details.

### 3.3 INFERENCE

We explore two strategies for defining the mapping function $\mathcal{M}$ during inference. Let $L$ represent the average token length per reasoning step, and $R$ the total number of steps in a reasoning trace.

**(1) Discretized (Language-Level)** Inspired by SentenceVAE (An et al., 2024), we apply a *decode-and-reencode* procedure: $\mathcal{M}(\hat{h}_t) = E(D(\hat{h}_t))$, where the predicted latent is first decoded into a sentence and then re-encoded into the model's input space. We refer to this as the **DISCRETIZED** mode, as each step explicitly traverses the discrete natural language interface. This approach helps mitigate error compounding (Simchowitz et al., 2025), but comes at a higher computational cost, with attention cost scaling as $\mathcal{O}(L^2 R + R^2)$. A detailed complexity analysis can be found in Appendix E.

**(2) Continuous (Latent-Level)** Following Coconut (Hao et al., 2024), we define the mapping as an identity function $\mathcal{M} = I$, directly propagating the predicted latent embedding $\hat{h}_t$ without intermediate decoding. In this **CONTINUOUS** mode, reasoning is entirely performed within the continuous embedding space, enabling significantly more efficient inference with attention complexity reduced to $\mathcal{O}(R^2)$.

Both methods offer computational advantages over natural language CoT, which incurs $\mathcal{O}(L^2 R^2)$ attention complexity even under key-value caching. However, the savings in the DISCRETIZED mode are conditional: they occur only when either (1) the encoder-decoder are not too computation-heavy, or (2) attention dominates over MLP cost—typially when the total output length $LR$ is relatively long (e.g., Blocksworld). Otherwise, the repeated decoding and encoding introduce additional **MLP overhead**.[4]

### 3.4 EXPERIMENTS

Building upon prior studies (Hao et al., 2024; Deng et al., 2024), we select GPT-2 as our baseline model and evaluate its performance across four distinct reasoning domains detailed in Section 2. To investigate optimal embedding strategies for latent reasoning, we examine **Semantic** and **Contextual** (both **Ctx-B** and **Ctx-C**) embeddings from Section 2. We further explore a hybrid architecture—**Sem (input)** → **Ctx (output)**—which mirrors the natural separation of representational roles found in conventional language modeling.

For evaluation, we compare sentence-level reasoning models against three baseline models. First, **CoT** represents a fully supervised model trained with access to both intermediate reasoning steps and final answers. Second, **No-CoT** omits step-level supervision and is trained solely to predict final answers. Third, we include **Coconut** (Hao et al., 2024), which gradually forgoes explicit token-level targets with curriculum-based substitution of fixed number last hidden states.

### 3.5 RESULTS

Our objective is to examine whether a latent sentence-level reasoning framework can generalize to higher-level abstractions while retaining the model's learned priors. Comparable performance to token-level Chain-of-Thought (CoT) would provide initial evidence, leading us to pose three research questions

**Q1: Can sentence-level reasoning match token-level CoT performance?** We hypothesize that effective reasoning is driven more by transitions between high-level concepts than by fine-grained token-level details. Empirically, sentence-level models match or even exceed CoT performance on logical and commonsense reasoning tasks. On mathematical and planning benchmarks, performance is slightly lower, though the gap remains modest. We attribute this to the greater precision often required in these domains, where continuous latent representations may be more prone to fidelity loss.

---

[4]Note that using a contextual encoder incurs greater computational cost than a semantic encoder.

| SETTING | PROSQA | CSQA | GSM8K | BLOCKSWORLD |
|---|---|---|---|---|
| **DIRECT** | | | | |
| **No-CoT** | 76.7 | 23.3 | 18.7 | 36.8 |
| **LANGUAGE-LEVEL** | | | | |
| **CoT** | 77.5 | 35.7 | 43.4 | 84.3 |
| *Sem.* | 83.6 | 28.5 | 38.9 | 32.9 |
| *Ctx-B.* | **91.4** | 35.2 | 39.0 | 70.0 |
| *Ctx-C.* | 79.8 | **40.3** | 37.1 | **76.3** |
| *Sem. → Ctx.* | 83.8 | 34.9 | **40.3** | 67.1 |
| **LATENT-LEVEL** | | | | |
| **Coconut** | 97.0 | 34.0 | 34.1 | 37.9 |
| *Sem.* | 86.0 | 27.5 | 29.6 | 30.8 |
| *Ctx-B.* | **92.6** | **37.0** | 37.4 | 70.5 |
| *Ctx-C.* | 81.6 | 35.5 | **38.3** | **80.8** |
| *Sem. → Ctx.* | 85.4 | 33.6 | 29.3 | 52.4 |

Table 2: Performance on PROSQA, CSQA, GSM8K, and BLOCKSWORLD across different embedding paradigms. **Bolded values** indicate the best performance **among our proposed methods** within each section. Baseline results are highlighted with background colors.

**Q2: How does sentence-level reasoning differ between language-level and latent-level inference?** To explore this, we compare model inference in the DISCRETIZED (language-level) space with that in the CONTINUOUS (latent-level) space. Results reveal complementary strengths: continuous models excel on logic and planning tasks, where reasoning benefits from uninterrupted latent-space composition and abstract state transitions. Conversely, discretized models show modest advantages on commonsense and mathematical benchmarks—likely due to the grounding effect of explicit linguistic representations. Still, the observed performance gaps are narrow—3.3% on commonsense and 0.7% on math—indicating that latent inference remains a viable and compute-efficient alternative. These findings suggest that effective reasoning need not always traverse explicit language space; continuous representations alone may support structured inference.

**Q3: Can sentence-level reasoning reduce computational cost?** Table 8 compares computational costs (FLOPs) between latent reasoning model and token-level CoT under forward-pass evaluation with key-value caching enabled. Latent reasoning employs an oracle answer classifier—executed via a single forward pass through the translator—that monitors the predicted embedding sequence and halts generation upon detecting a special answer token. The final latent embedding is decoded into natural language for evaluation.

Table 3: Average inference-time compute cost (GFLOPs) for each dataset under CoT and CTX-C CONTINUOUS Inference.

| DATASET | CoT | CTX-C |
|---|---|---|
| CSQA | 25.89 | 9.96 |
| PROSQA | 100.99 | 70.19 |
| GSM8K | 21.45 | 12.68 |
| BLOCKSWORLD | 58.69 | 28.57 |

Note that we measure computational costs across the full latent pipeline, including classifier and decoder components, which remain unoptimized.[5] Thus, reported efficiency gains represent conservative estimate. Across tasks, CONTINUOUS inference achieves 1.5–2.5× better efficiency compared to token-level CoT. Notably, we highlight that even DISCRETIZED inference outperform CoT in longer reasoning tasks (e.g., Blocksworld w/ average trace length $R \sim 9.1$: 52.26 GFLOPs vs. 58.69 GFLOPs). We expect this efficiency gap to grow as the length of reasoning trace increases.

---

[5]To see the cost with a lightweight classifier, please refer to Appendix F.

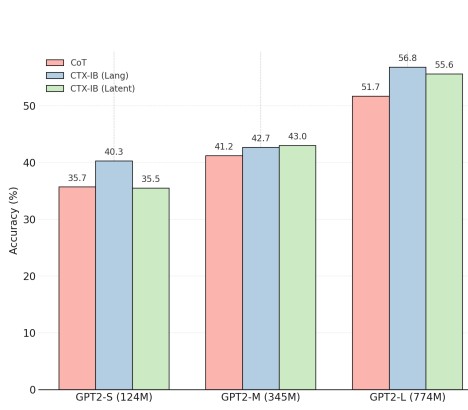

Figure 3: CoT vs. CTX-C on CommonsenseQA across GPT-2 variants.

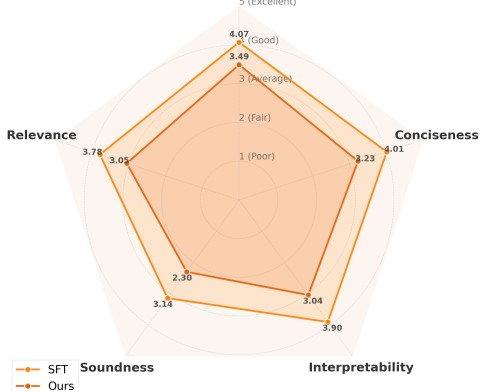

Figure 4: GPT-4o qualitative evaluation of the reasoning steps evaluated using a similar metric employed in (Ye et al., 2023), where SFT is trained using CoT and ours is using CTX-B.

## 4 DISCUSSION

### 4.1 POTENTIAL SCALABILITY AND MODULARITY

**Scalability** We report preliminary observations that suggest our framework has potential to scale to increasing model capacity. Due to computational constraints, our experiments are limited to sub-1B models; we evaluate GPT-2 Medium (345M) and GPT-2 Large (775M) on the CommonsenseQA (CSQA) benchmark, which exhibits clear performance scaling under CoT fine-tuning. As shown in Figure 3, the **Ctx-C** configuration attains performance comparable to, and in some cases exceeding, CoT—despite operating entirely in latent space and incurring lower inference-time compute. While tentative, these findings suggest that latent reasoning could offer a more compute-efficient path toward generalization.

**Using Off-the-Shelf Encoder–Decoder** We investigate whether the encoder–decoder can be decoupled from the latent model and replaced with smaller, fixed components. This modular design seeks to reduce the computational burden of DISCRETIZED inference—especially in settings where only the latent reasoning module requires adaptation. To evaluate this hypothesis, we paired a lightweight GPT-2 Small encoder–decoder (trained on Ctx-C) with a GPT-2 Medium latent model and assessed performance on GSM8K.[6]

This hybrid configuration achieved an accuracy of **42.23**, compared to **47.69** for a fully fine-tuned GPT-2 Medium with CoT training. While accuracy decreases slightly, the results demonstrate that predictive embeddings can transfer across model architectures with reasonable degradation–supporing the feasibility of modular reuse. Given prior findings on general embedding space alignment across models (Conneau et al., 2018; Jha et al., 2025), further exploration with larger models and diverse tasks remains a promising direction.

### 4.2 SENTENCELENS: TOWARDS *Human-Readable* INTERPRETABILITY

We introduce **SentenceLens**, an intrepretability tool that decodes intermediate hidden representations by directly passing them through the trained sentence-level decoder. In contrast to token-level inspection methods such as Logit Lens (nostalgebraist, 2020), SentenceLens operates at the sentence level, offering **a more human-readable view** of the model's evolving internal states across reasoning steps.

---

[6]GSM8K was selected based on preliminary findings that moderately sized datasets help stabilize shallow MLP mappings across heterogeneous embedding spaces.

For example, in Table 6, we show how the model's prediction shifts across layers during the transition from one reasoning step to the next. When making first step prediction $\hat{h}_1$, Layer 19 introduces a general observation about eating and energy levels, while Layer 22 begins to center on the idea that hunger motivates goal-directed behavior. These intermediate activations reflect a gradual shift in conceptual focus, which in the last layer ($36^{th}$) develops as: ***If you are hungry, you are likely engaging in an activity that requires sustenance.*** Since the latent model frames reasoning as a *continuous process*, we hypothesize that intermediate latent states may become naturally decodable—allowing us to observe the progression of inference across steps. To see more examples, see Appendix A.

**Qualitative Analysis** In addition, when decoding output embeddings at successive latent reasoning steps (*e.g.*, Step 1 through Step 5), we find that the resulting sentences, while readily understandable, often lack the coherence and rigor characteristic of standard CoT responses. We compare two model outputs using GPT-4o evaluation with the rubric proposed by Ye et al. (2023). This scores Relevance, Fluency, Conciseness, Soundness, and Interpretability on a 1 to 5 Likert scale. It turns out that CTX-C model mostly produces reasoning chains of moderate quality (scores ¿ 3); However, its performance falls short compared to CoT models trained directly in natural language space (Figure 4). The largest weakness appears in Soundness, which aligns with earlier observations that high-level concept models may exhibit reduced coherence even after extensive pretraining (team et al., 2024). While we believe this tradeoff is a natural consequence of abstraction, bridging this gap remains an interesting direction for future research.

## 5 RELATED WORKS

**Sentence Representations and Prediction** Sentence-level representation learning has historically followed two main paradigms: *reconstruction* and *context prediction*. Early methods, such as sequence autoencoders (Dai & Le, 2015) and Skip-Thought vectors (Kiros et al., 2015), learned embeddings by reconstructing input or neighboring sentences. More recent approaches move beyond token-level generation to predict entire sentences, including latent-variable models such as VAEs (Bowman et al., 2016) and hierarchical decoders (Serban et al., 2016), as well as LCM (team et al., 2024) and CoCoMix (Tack et al., 2025). Building on these developments, our framework defines semantic and contextual embeddings, employs contrastive learning to align latent input-output pairs (van den Oord et al., 2019), and distinguishes itself by leveraging pretrained models to introduce latent reasoning mechanisms rather than training from scratch.

**Latent-Space Reasoning** Efficiency and abstraction have motivated reasoning directly in embedding space, bypassing token generation. Joint embedding architectures (Assran et al., 2023) and predictive coding frameworks (van den Oord et al., 2019) model representation dynamics by forecasting future embeddings. This idea has recently been extended to language: Hao et al. (2024) introduced *continuous latent reasoning*, where token-level embeddings are gradually replaced with continuous embeddings with the last-layer hidden states through a curriculum-based strategy from Deng et al. (2024).

## 6 CONCLUSION

We present a framework that elevates pretrained language models from token-level generation to sentence-level reasoning by autoregressively predicting continuous embeddings of next-step sentences. This enables reasoning over more abstract conceptual units while retaining pretrained inductive biases. Our exploration of semantic and contextual embeddings reveals that contextual embeddings show competitive performance with token-level Chain-of-Thought (CoT) across diverse reasoning tasks, while significantly reducing inference-time computational costs under Continuous inference. Additionally, we demonstrate signs of scalability, modular reuse of encoder–decoder components, and enhanced interpretability through SentenceLens, which decodes latent embeddings into human-readable sentence-level traces. These findings suggest that pretrained language models could be effectively adapted for structured reasoning in latent embedding spaces, opening new directions for efficient latent reasoning systems.

REPRODUCIBILITY STATEMENT

We provide all codes for dataset generation, training, and analysis in the Appendix and the an-noymized repository.

LLM USAGE DISCLOSURE

We adopted LLMs to fix grammatical errors and refine any unnatural expressions.

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

## A SENTENCELENS EXAMPLES

We include a representative **SentenceLens** example that highlights additional key observations. Specifically, the model often identifies the correct answer early in the latent trajectory; however, subsequent chain-of-thought (CoT) tokens exhibit a drift that ultimately leads to an incorrect prediction. (The correct answer is **C**.) This suggests room for improvement by using intermediate representations as explicit supervision targets, which could guide the construction of model centric datasets and self-training methods.

Table 4: **Example of Latent Reasoning Trajectory inspected with SentenceLens.** Although early steps' intermediate layers demonstrate accurate associations with hair loss and balding, the final prediction selects an incorrect choice, showing a drift in reasoning at later stages.

| Step | Decoded Sentence(s) |
|---|---|
| Question | *For many males hair is a concern as they get older, it begins to what, causing a receding hairline?* 
 **A: thin out     B: grow in ear     C: fall out     D: bulge     E: composted** |
| $0 \rightarrow 1$ | LAYER 19: The human body requires a certain amount of energy to maintain its functions. 
 LAYER 20: The primary cause of aging is the loss of moisture. |
| 1 | *One of the common changes in hair density over time is the decrease in hair volume.* |
| $1 \rightarrow 2$ | LAYER 4: A common reason for hair loss is due to a decrease in hair density and diameter. 
 LAYER 23: The aging process causes various health issues. |
| 2 | *The hair loss is often associated with hair loss.* |
| $2 \rightarrow 3$ | LAYER 3: This process is often referred to as balding. 
 LAYER 23: The aging process leads to reduced body size. |
| 3 | *A thinning hairline is commonly associated with hair loss.* |
| $3 \rightarrow 4$ | LAYER 2: This process can lead to a decrease in hair density and diameter. 
 LAYER 11: The process of getting older leads to the body becoming thinner. |
| 4 | *This process is commonly referred to as fading.* |
| $4 \rightarrow 5$ | LAYER 4: This process is common in older individuals who lack regular hair growth. |
| 5 | *### A (Incorrect)* |

Table 5: **Early Answer Emergence in Latent Reasoning.** The model brings up the concept of "commuting" in the reasoning chain even before the first autoregressive step completes. This hints at potential efficiency gains by leveraging early, confident predictions as supervision signals in training.

| Step | Decoded Sentence(s) |
|---|---|
| Question | *Why would you take a bus to work?* 
 **A: commute    B: flying    C: get somewhere    D: travel    E: go home** |
| **0 → 1** | LAYER 19: A person spends time traveling between different locations. 
 LAYER 20: A person spends time commuting to work. 
 LAYER 21: A person spends time traveling, which often involves moving from one place to another. 
 LAYER 22: A person spends time traveling, which often involves traveling across distances. |
| **1** | *People often take the bus to reach a destination.* |
| | *... not shown* |
| **5** | *### A (Correct)* |

Table 6: **Latent Sentence Transitions with SENTENCELENS** for GPT2-Large under the CTX-C, CONTINU-OUS setting. We visualize intermediate decoding across layers and reasoning steps. Highlighted rows represent the output from the final latent embedding at each step.

| Step | Decoded Sentence(s) |
|---|---|
| Question | *If you are hungry and going fishing, why would you be going fishing?* 
 **A: to see the fish    B: have fun    C: catching fish    D: wet clothes    E: killing** |
| **0 → 1** | LAYER 19: A person who eats a lot experiences increased energy levels. 
 LAYER 22: A person who is hungry seeks to alleviate their hunger. When you are hungry, you engage in an activity to satisfy your hunger. ⋯ |
| **1** | *If you are hungry, you are likely engaging in an activity that requires sustenance.* |
| **1 → 2** | LAYER 9: If a person is hungry, they are likely to engage in eating. 
 LAYER 20: The act of catching fish involves physical activity. |
| **2** | *Fishing is a common activity for those who enjoy the outdoors.* |
| **2 → 3** | LAYER 4: Fishing is a common activity for those who enjoy catching fish. 
 LAYER 21: The act of catching fish can lead to enjoyment and recreation. |
| **3** | *Fishing is a recreational activity that people engage in for fun.* |
| **3 → 4** | LAYER 9: The act of catching fish provides a direct source of food. 
 LAYER 21: The act of catching fish provides a direct source of food. People fish to enjoy the experience of catching fish. |
| **4** | *Fishing is a recreational activity that people often engage in.* |
| **4 → 5** | LAYER 5: Fishing is a recreational activity that is often pursued with friends. Therefore, fishing is a good reason to go fishing. |
| **5** | *### C* |

## B    FRAGILITY OF CONTINUOUS EMBEDDINGS

Latent reasoning operates over high-dimensional embedding manifolds, which tend to be more sensitive to perturbations than discrete token-level autoregression (team et al., 2024; Simchowitz et al., 2025). To systematically assess this *fragility*, we introduce synthetic noise at inference time, following team et al. (2024) with a 50% probability. We evaluate robustness across three intervention points in the reasoning pipeline: (1) Language-Level (Input): noise is applied to the input embedding; (2) Language-Level (Output): noise is added to the output embedding; and (3) Latent-Level: noise is directly injected into the predicted output embedding, which is then autoregressively consumed in the next step.

Table 7: **Natural Language CoT Trace**. Output from the CoT trained model (CoT)

| **CoT Model Reasoning Trace** |
|---|
| If you are hungry, you likely seek food to satisfy that hunger. |
| Fishing is an activity that typically results in catching fish. |
| Catching fish is a common reason for going fishing. |
| Seeing the fish is a primary motivation for engaging in fishing. |
| **### C** |

Empirically, we observe two key trends: (1) performance degrades more rapidly on GSM8K, where precise numerical reasoning amplifies the impact of noise; and (2) *Language-Level* inference (*i.e.,* decoding and re-encoding) consistently yields greater robustness than latent-only reasoning across both tasks. This supports the intuition that grounding in language acts as a regularizing prior, mitigating error accumulation at the cost of additional compute. These findings highlight a trade-off between efficiency and stability, motivating future work on approaches that help prevent error compounding.

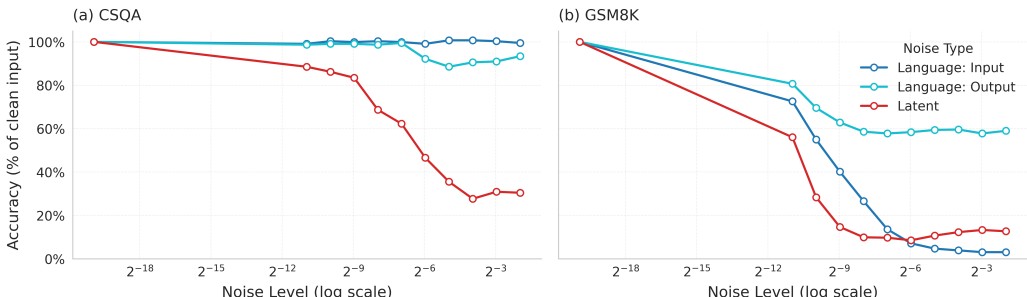

Figure 5: Performance Change when injecting a Gaussian random noise to different modes of inference, for Ctx-C model in GSM8K and CSQA datasets.

## C DATASET DESCRIPTION

**Mathematics**   We use the GSM8K dataset (Cobbe et al., 2021), which consists of grade-school math word problems originally comprising 7.8k training and 1.3k test samples. Following prior expansions (Hao et al., 2024; Deng et al., 2024), we adopt an extended version containing approximately 370k training examples to support large-scale latent model training.

**Planning**   Following prior work (Bohnet et al., 2024), we use the Blocksworld environment for planning evaluation, but construct the dataset generation pipeline using our own Python implementation. We evaluate the model on 7-block configurations, ensuring that the initial and goal states do not overlap across the training, validation, and test sets. We use 9.9k samples for training, and 380 samples each for testing.

**Logical**   We adopt ProsQA (Hao et al., 2024), a synthetic dataset grounded in first-order logic. Each instance presents multiple distractors and requires multi-hop reasoning over a structured graph. Prior work highlights that latent models capable of multi-state tracking exhibit strong performance on this task. We use a 17.8k training set and 500 samples for evaluation.

**Commonsense**   We use CommonsenseQA (Talmor et al., 2019), a multiple-choice benchmark that lacks explicit Chain-of-Thought (CoT) supervision. To enable training with intermediate reasoning steps, we augment the data using GPT-4o to generate CoT-style rationales. Our training split includes 8.5k examples, and for evaluation, we reserve 611 samples from the validation set.

Figure 7 illustrates representative examples from each dataset.

## D LIMITATIONS

**Fragility of Latent Reasoning**   As illustrated in Figure 5, pure latent reasoning, as it is conducted entirely within a continuous embedding space, becomes notably fragile. Unlike DISCRETE-STEP inference, which introduces a discrete decoding step that inherently quantizes minor perturbations, the continuous pathway lacks such built-in stabilization. This discrete bottleneck serves as a form of regularization, filtering out numerical noise and constraining the model's trajectory to a finite set of linguistically meaningful sequences. To mitigate this, specialized stabilization mechanisms are required (Simchowitz et al., 2025), and future work could explore hybrid frameworks that incorporate discrete bottlenecks at critical points in the reasoning process.

**Training from Scratch**   Training a model from scratch directly in the higher abstractions *i.e. sentence embeddings* space appears to be a straightforward path toward robust high-level reasoning. Prior work argues that models initialized on discrete-token objectives must later overcome a distribution shift when asked to operate over sentence-level abstractions, and this difficulty intensifies as model size—and pretraining data size—increase (Hao et al., 2024; Deng et al., 2024) leading subsequent work to favor pretraining" (Goyal et al., 2024; Shen et al., 2025)

However our adaptation framework shows that a pretrained token-level language model can be lifted, with modest additional supervision, onto an interpretable sentence-manifold without retraining everything from scratch. By demonstrating both the promise and the fragility of this approach, the present work highlights a critical research frontier: designing models that learn to abstract while preserving previously learned inductive bias.

# E  COMPUTATION COMPLEXITY ANALYSIS

**Attention Complexity under KV-caching**  Let $L$ be the average number of tokens per sentence, $R$ the number of reasoning steps, and ignore the prompt length $N_0$ in leading order.

**(1) Chain-of-Thought (CoT).** Each step emits $L$ new tokens into the context. Before step $t$, the context length is $N_0 + (t-1)L$, so

$$\mathcal{C}_{\text{CoT}} = \sum_{t=1}^{R}\sum_{j=1}^{L}\big[N_0 + (t-1)L + (j-1)\big] = \mathcal{O}\big(L^2 R^2\big).$$

**(2) Contextual Embedding Mode.** At each step the model (i) decodes one latent into an $L$-token sentence and (ii) attends over all retained tokens to predict the next latent:

$$\underbrace{\sum_{t=1}^{R}\sum_{j=1}^{L} j}_{\mathcal{O}(L^2 R)} + \underbrace{\sum_{t=1}^{R}(N_0 + (t-1)L)}_{\mathcal{O}(L\,R^2)} = \mathcal{O}\big(L^2 R + L\,R^2\big).$$

**(3) Language-Grounded Mode.** Each step (i) processes only latents in the main chain $\big(\mathcal{O}(R^2)\big)$ and (ii) decodes and re-encodes an $L$-token sentence $\big(\mathcal{O}(L^2 R)\big)$, yielding

$$\mathcal{C}_{\text{LG}} = \mathcal{O}\big(R^2 + L^2 R\big).$$

**(4) Pure Latent Mode.** Each step adds one latent vector; attending over $t-1$ latents gives

$$\mathcal{C}_{\text{latent}} = \sum_{t=1}^{R}(N_0 + t - 1) = \mathcal{O}\big(R^2\big).$$

**Summary of leading-order costs:**

$$\mathcal{C}_{\text{CoT}} = O(L^2 R^2), \quad \mathcal{C}_{\text{contextual}} = O(L^2 R + L\,R^2), \quad \mathcal{C}_{\text{LG}} = O(L^2 R + R^2), \quad \mathcal{C}_{\text{latent}} = O(R^2).$$

**MLP Overhead**  In addition to attention cost, every decoded or re-encoded token incurs feed-forward (MLP) computation. More specifically:

- **CoT & Contextual Embedding:** emits $L$ tokens per step $\rightarrow$ processes $L \times R$ tokens through MLP $\rightarrow \mathcal{O}(LR)$.
- **Language-Grounded:** With a *semantic* encoder, each step decodes and re-encodes $L$ tokens on compact codes—processing $2L$ tokens per step for an MLP cost of $\mathcal{O}(LR)$. If instead a *contextual* encoder must re-attend over up to $N_0 + (t-1)L$ tokens each pass, it incurs an additional $\mathcal{O}(LR^2)$ MLP overhead, which can erode attention savings unless the encoder is shallow or non-autoregressive.
- **Pure Latent:** processes one latent per step $\rightarrow \mathcal{O}(R)$.

**Concluding Remark**  Under KV-caching, the *Language-Grounded* mode—with a **semantic** encoder—adds an $\mathcal{O}(L^2 R)$ decode/re-encode overhead, but makes it ideal for tasks sensitive to error-compounding or instability (*i.e.* Mathematics.) In contrast, the *Pure Latent* mode eliminates all token-level context (attention $\mathcal{O}(R^2)$, MLP $\mathcal{O}(R)$), offering maximal efficiency when possible.

## F  TERMINATION CLASSIFIER

While we initially assume an oracle termination signal by using the first token generated by the decoder, we also demonstrate that this decision can be learned by a lightweight classifier. Specifically, we train a three-layer feedforward neural network (MLP) to identify the *answer* sentence during CONTINUOUS inference. The MLP consists of linear layers with hidden dimensions of 192 and 48, each followed by a GELU activation, and outputs a single logit for binary classification (continue versus terminate). It is trained using binary cross-entropy loss with logits (BCEWithLogitsLoss). Note that the average inference GFLOPs, reported in Table 9, are lower than those reported in Table 3.

## G  EXPERIMENT DETAILS

Each dataset requires task-specific hyperparameter choices due to variation in problem structure and reasoning complexity. For all experiments, we report the best test-set accuracy across saved checkpoints (including baselines). When training all of our models (Latent Model, Encoder, and Decoder), we initialize them from the SFT checkpoint. The number of training epochs for each stage was selected based on convergence trends observed during early stage of experiments. Please note that we use small portion of Fineweb-Edu (Penedo et al., 2024) for CSQA task's restoration (*i.e.* training for *semantic* embeddings.) We report hyperparameters used in Table 10 and 11.

## H  EVALUATION PROMPT

Please refer to Figure 6.

Table 8: Dataset statistics for each reasoning benchmark across train, validation, and test splits.

| Split | Metric | CSQA | ProsQA | GSM8K | Blocksworld |
|-------|--------|------|--------|-------|-------------|
| TRAIN | Question tokens/sample | 39.0 | 360.4 | 42.2 | 146.8 |
|       | Steps/sample | 5.6 | 3.8 | 3.6 | 8.9 |
|       | Tokens/step | 10.9 | 9.5 | 6.0 | 8.0 |
| VALID | Question tokens/sample | 38.4 | 361.0 | 55.1 | 146.5 |
|       | Steps/sample | 5.6 | 3.8 | 4.2 | 9.2 |
|       | Tokens/step | 10.7 | 9.5 | 6.0 | 8.0 |
| TEST  | Question tokens/sample | 38.8 | 357.0 | 56.8 | 146.6 |
|       | Steps/sample | 5.6 | 3.8 | 4.3 | 9.1 |
|       | Tokens/step | 10.8 | 9.5 | 6.1 | 8.0 |

Table 9: Average inference-time compute cost (GFLOPs) on each dataset under CoT and CTX-C CONTINUOUS inference, with the accuracy of the trained classifier.

| DATASET | CoT | CTX-C | CLASSIFIER ACCURACY |
|---------|-----|-------|---------------------|
| CSQA | 25.89 | 8.51 | 99.36 |
| PROSQA | 100.99 | 64.02 | 99.76 |
| GSM8K | 21.45 | 10.80 | 99.46 |
| BLOCKSWORLD | 58.69 | 26.73 | 97.95 |

Table 10: Training configurations of GPT-2 for each dataset and training stage. *SFT includes both CoT and No-CoT variants.

| Stage | GSM8K | CSQA | ProsQA | Blocksworld |
|---|---|---|---|---|
| **SFT*** | | | | |
| Epochs | 20 | 20 | 20 | 100 |
| LR | 1e-4 | 1e-4 | 1e-4 | 1e-4 |
| Batch | 64 | 64 | 64 | 64 |
| **EMBEDDING: RESTORATION** | | | | |
| Epochs | 3 | 5 | 3 | 100 |
| LR | 5e-4 | 5e-4 | 5e-4 | 1e-4 |
| Batch | 256 | 512 | 128 | 1024(256*4) |
| **EMBEDDING: PREDICTION** | | | | |
| Epochs | 30 | 50 | 50 | 50 |
| LR | 5e-4 | 5e-4 | 5e-4 | 1e-4 |
| Batch | 128 | 128 | 96 | 64 |
| **LATENT AUTOREG.** | | | | |
| Epochs | 200 | 300 | 50 | 200 |
| Eval Freq | every 10 | every 10 | every 2 | every 10 |
| LR | 5e-4 | 5e-4 | 5e-4 | 5e-4 |
| Batch | 128 | 128 | 32 | 64 |

Table 11: Training configurations by model size and stage. LoRA configuration used for GPT-2 Large.

| Stage | GPT-2 Small | GPT-2 Medium | GPT-2 Large (LoRA) $r$=256, $a$=1024) |
|---|---|---|---|
| **SFT** | | | |
| Epochs | 20 | 20 | 20 |
| LR | 1e-4 | 1e-4 | 1e-4 |
| Batch | 64 | 64 × 8 | 64 × 8 |
| **EMBEDDING: RESTORATION** | | | |
| Epochs | 5 | 5 | 5 |
| LR | 5e-4 | 5e-4 | 5e-4 |
| Batch | 512 | 128 | 128 |
| Notes | used FW subset | used FW subset | used FW subset |
| **EMBEDDING: PREDICTION** | | | |
| Epochs | 50 | 50 | 50 |
| LR | 5e-4 | 5e-5 | 1e-4 |
| Batch | 128 | 128 | 64 |
| **LATENT AUTOREG.** | | | |
| Epochs | 300 | 300 | 300 |
| Eval Freq | every 10 | every 10 | every 2 |
| LR | 5e-4 | 1e-4 | 1e-4 |
| Batch | 128 | 64 | 128 |
| Notes | — | — | w. grad ckpting |

```
You are an impartial and rigorous evaluator.
Below is a multiple-choice question and two reasoning traces produced by two
anonymised models (model1 and model2).  A reference answer is provided, but
your task is to evaluate the **quality of the reasoning**, *independent* of
whether the final answer is correct.

### Evaluation Rubric (1 = very poor, 2 = poor, 3 = average, 4 = good, 5 = excellent)

1. **Fluency** – Is the text grammatically correct and easy to read?
2. **Relevance** – Does it meaningfully address the question or topic?
3. **Soundness** – Is the reasoning logically valid and/or factually accurate?
4. **Interpretability** – Can a human clearly understand what is being argued?
5. **Conciseness** – Is the point expressed efficiently without redundancy?

### Output format

Return **only** a JSON object with **exactly** the following structure:

{
  "fluency":        {"model1": 1-5, "model2": 1-5},
  "relevance":      {"model1": 1-5, "model2": 1-5},
  "soundness":      {"model1": 1-5, "model2": 1-5},
  "interpretability":{"model1": 1-5, "model2": 1-5},
  "conciseness":    {"model1": 1-5, "model2": 1-5}
}
```

Figure 6: Evaluation Prompt used to GPT-4o for judging intermediate reasoning step's quality.

**GSM8K**

Janet's ducks lay 16 eggs per day. She eats three for breakfast every morning and bakes muffins for her friends every day with four. She sells the remainder at the farmers' market daily for $2 per fresh duck egg. How much in dollars does she make every day at the farmers' market?

<<16-3-4=9>>
<<9*2=18>>
18

**ProsQA**

Every gerpus is a terpus. Every terpus is a zhorpus. Every lempus is a yerpus. Every boompus is a zhorpus. Every brimpus is a rempus. Every lempus is a jelpus. Every lorpus is a rorpus. Bob is a yerpus. Every worpus is a rempus. Every lempus is a impus. Every rempus is a sterpus. Every yimpus is a zumpus. Every lempus is a yumpus. Every shumpus is a jelpus. Every brimpus is a zhorpus. Every scrompus is a rempus. Every lempus is a wumpus. Sally is a boompus. Sally is a gerpus. Every gerpus is a scrompus. Bob is a wumpus. Every wumpus is a lorpus. Every yerpus is a rorpus. Sally is a terpus. Every gerpus is a zhorpus. Every yimpus is a zhorpus. Every boompus is a terpus. Every gerpus is a worpus. Bob is a lorpus. Every gerpus is a yimpus. Every scrompus is a brimpus. Every lempus is a rorpus. Every lempus is a shumpus. Bob is a jelpus. Sally is a scrompus. Every gerpus is a brimpus. Every lempus is a lorpus. Every boompus is a yumpus. Every scrompus is a zumpus. Every zhorpus is a tumpus. Sally is a zumpus. Every lempus is a storpus. Every yerpus is a lorpus. Every scrompus is a zhorpus. Every yimpus is a rempus. Every impus is a jelpus. Jack is a yimpus. Every yerpus is a wumpus. Every rorpus is a hilpus. Every yimpus is a sterpus. Bob is a lempus. Every worpus is a storpus. Every rorpus is a impus. Every boompus is a gerpus. Is Sally a hilpus or sterpus?

Sally is a scrompus
Every scrompus is a rempus.
Every rempus is a sterpus.
### Sally is a sterpus.

**CSQA**

They were kissing each other good bye, they had no worries because their relationship had a strong foundation of what?
A: partner   B: trust   C: cooperation   D: bricks   E: herpes

Trust is fundamental to a strong relationship.
Relationships with strong foundations typically rely on trust.
Trust allows partners to feel secure in their relationship.
Saying goodbye without worries indicates a high level of trust.
### B

**Blocksworld**

In the Blocksworld domain, blocks can be stacked on top of each other or placed on the table. Only one block can be moved at a time, and only clear (unblocked) blocks can be moved. Given the initial configuration and the goal configuration, what is the minimum number of moves required to reach the goal? Initial state: A is on the table, B is on A, C is on G, D is on F, E is on B, F is on the table, G is on E. Goal state: A is on the table, B is on the table, C is on D, D is on B, E is on the table, F is on the table, G is on E.

<<Move C from G to the table>>
<<Move G from E to the table>>
<<Move E from B to the table>>
<<Move B from A to the table>>
<<Move D from F to B>>
<<Move C from the table to D>>
<<Move G from the table to E>>
The answer is: 7

Figure 7: Example instances from each dataset.

