# OpenReview forum: "Latent Reasoning via Sentence Embedding Prediction"
_ICLR.cc/2026/Conference — Submitted to ICLR 2026_

### Official Review · Reviewer_Ncgs · 2025-10-28

**Soundness:** 4
**Presentation:** 3
**Contribution:** 3
**Rating:** 4
**Confidence:** 3

**Summary:**

This paper proposes a novel framework that elevates pretrained language models from token-level generation to sentence-level reasoning. Instead of predicting tokens, the model autoregressively predicts continuous embeddings of the next sentence, supporting two embedding paradigms—semantic and contextual. The authors further design two inference modes: discretized and continuous. Experiments show that contextual embeddings under continuous inference achieve competitive performance with CoT reasoning while halving inference FLOPs.

**Strengths:**

1-The paper takes a new step beyond token-level CoT reasoning by framing reasoning as prediction over sentence embeddings.

2-The experiments span mathematical, logical, commonsense, and planning tasks, showing consistent findings and careful analysis of efficiency and robustness.

**Weaknesses:**

1-Although the idea is intriguing, the improvements over CoT are modest and not statistically analyzed; performance is often close to, but not consistently better than, token-level baselines.

2-All experiments are limited to sub-1B GPT-2 models; the claimed scalability to larger models is only hypothesized.

3-SentenceLens is interesting but lacks quantitative validation (e.g., measuring interpretability gains or correlation with reasoning correctness).

4-The paper contains many technical details and citations but could be clearer.

5-The reported performance gains may largely stem from fine-tuning effects rather than genuine enhancements in reasoning capability. The proposed method’s contribution to reasoning improvement is therefore not convincingly demonstrated.

6-Several comparative algorithms and components are mentioned, yet the paper lacks rigorous ablation or empirical evidence showing their individual effectiveness or contribution to the final results.

7-As stated on page 5, it remains unclear whether the observed reasoning ability truly originates from the latent model itself, or from the pretrained transformer’s inherited capability.

8-While SentenceLens offers interpretability, it requires an additional decoding module and increases computational complexity. Thus, the interpretability improvement comes at the cost of additional model overhead rather than being an inherent property of the latent reasoning framework.

**Questions:**

1-Can the proposed latent reasoning generalize to open-ended generation tasks (e.g., summarization or dialogue) rather than step-bounded reasoning?

2-How does the approach compare to recent reasoning compression methods?

---

> ### Author Response · Authors · 2025-12-01
>
> Dear Reviewer, thank you for your constructive review.
>
> While the review system does not allow us for further communication, below we address your concerns to our best effort.
>
> ---
>
> **W) 1,5: Improvements over CoT are modest; may come from fine‑tuning rather than reasoning.**
>
> *Regarding W1:*
>
> Our goal is parity with CoT at markedly lower inference compute, and not **necessarily to dominate the token‑level CoT on every benchmark.** In other words, we aim to provide a preliminary study on whether latent reasoning with sentence embedding could come close to the performance of a model fine-tuned to output CoT reasoning token by token.
>
> In Table 2, continuous latent models (Latent‑Level) are typically within a few points of CoT on GSM8K/CSQA/Blocksworld, and substantially stronger on ProsQA when using the unregularized contextual variant (ProsQA: Ctx‑B 92.6 vs CoT 77.5). At the same time, Table 3 shows 1.5–2.5× lower GFLOPs for CTX‑C (Continuous) than CoT, measured end‑to‑end and including the translator and classifier. These results support our central claim, which is about efficiency under similar performance.
>
> *Regarding W5:*
>
> To address the “fine‑tuning effect” concern, note that all models share the same SFT backbone before we introduce sentence‑level training. (This means that every method in Table 2 is fine-tuned including all baselines). The delta from SFT therefore reflects the effect of the latent sentence‑prediction objective and inference mode, as all our **Comparative baselines involve fine-tuning.** Especially,  (i) No‑CoT (Direct) shows  significantly poor accuracy despite being fine‑tuned (e.g., CSQA 23.3; GSM8K 18.7), indicating that fine‑tuning alone (without reasoning) does not recover the gains without structured intermediate steps.
>
> ---
>
> **W) 3, 8 SentenceLens lacks quantitative validation; it adds overhead.**
>
> SentenceLens is a **diagnostic**: it decodes **intermediate latent states** with the same trained sentence-level decoder. We extend the line of Prior analytical tools such as Logit Lens are mostly used qualitatively; yet we agree that adding quantitative probes in future would further strengthen the case.
>
> *On overhead*: if by “overhead” you mean decoding **last hidden states**, note that this is **not required at inference in the Continuous mode** for intermediate tokens. In most deployments we seek the accuracy boost of CoT-style reasoning but only need the **final answer**, so we do **not** decode intermediate tokens. Even if one *does* choose to inspect them, the cost remains below conventional token-level autoregressive decoding. Please refer to Appendix C for more details.
>
> ---
>
> **W6) Missing ablations for components.**
>
> At our best ,we have provided ablation about our core framework: (i) the formation of sentence-level embeddings (Sem, CTX-B/CTX-C) and (ii) the latent transition model trained on these embeddings, including both Continuous and Discretized inference (Sections 2–3). We also test modularity by swapping in an off-the-shelf encoder–decoder while adapting only the latent module, showing that the latent component carries the reasoning behavior.
>
> We agree additional ablations on our framework (i.e. loss design) could be beneficial , yet we want to highlight again that our primary aim, however, is to provide preliminary evidence that pretrained models can be lifted into an abstract reasoning space using their own compressed embeddings, rather than to claim our method’s superiority over alternatives.
>
> ---
>
> **W7) Is the reasoning in the latent model or just the pretrained transformer?**
>
> Our latent module is a post-training adaptor over an SFT backbone trained with next-token prediction (NTP). It inherits the pretrained/SFT priors but learns its own transition policy in sentence-embedding space (SEM / CTX-B / CTX-C from §2) while the encoder–decoder remain frozen during latent training (§3). At inference, the Continuous rollout executes this learned policy directly in latent space.
>
> In summary, our position is not that the latent module invents reasoning from scratch, nor that the backbone alone explains the gains: the SFT backbone provides the NTP-learned priors, but the latent transition is causally necessary, converting those priors into structured multi-step reasoning. The modularity swap (replacing the encoder–decoder while adapting only the latent model) preserves performance, and No-CoT remains weak under the same SFT. Altogether, this indicates that the improvement is due to the latent transition adapting pretrained priors; the backbone is necessary but not sufficient.

---

> > ### Author Response · Authors · 2025-12-01
> >
> > **Q1) Can the proposed latent reasoning generalize to open-ended generation tasks (e.g., summarization or dialogue) rather than step-bounded reasoning?**
> >
> > Yes, our framework naturally extends to open-ended outputs by training the termination head (or a lightweight latent policy) to detect “end-of-summary/turn-complete” states; at inference, Continuous mode would roll latent sentences, stop, then render the final (or batched) sentences to text.
> >
> > For open-ended tasks, the model would have to ultimately render each output sentence, so the compute savings are smaller than in step-bounded settings where we decode only the final answer. Still, because planning occurs in latent space (rather than token-by-token CoT), the end-to-end cost remains lower.
> >
> > There is a strong evidence that this style of latent planning works in open-ended generation. This is already shown by our strong results in CSQA and also was observed in Large Concept Models (LCM) [1], which predict sequences of sentence representations in the SONAR space and then decode to text. Here, they report strong results on summarization and a new summary-expansion task. This mirrors our design and suggests benefits of sentence-level generation for long-form structure on open-ended generation tasks.
> >
> > ---
> >
> > **Q2) How does the approach compare to recent reasoning compression methods?**
> >
> > Our method emphasizes three strengths:
> >
> > - (1) we can decode intermediate latent states with the same sentence-level decoder (SentenceLens) for human-readable trajectories;
> >
> > - (2) Our method uniquely supports token-level sampling during latent-level reasoning, opening exciting research avenues such as self-training and reinforcement learning
> >
> > - (3) training simplicity/scalability: unlike iterative latent curricula, we train in a single forward pass and, under Continuous inference, report substantial FLOPs savings vs. CoT.
> >
> > In this light, to compare with other reasoning compression methods:
> >
> > - CoDi [2] : Uses teacher–student distillation to build a latent reasoning model, with a fixed number of latent reasoning tokens; this design is task-specific and still implies multiple forward passes during training.
> >
> > - Token-Assorted [3] :  Replaces early CoT steps with VQ-VAE discrete latent tokens and mixes latent + text tokens to shorten traces, improving accuracy and reducing trace length. However, it doesn’t transition everything into latent space; instead it introduces extra latent tokens that aid reasoning, with a separate codebook/decoder.
> >
> > - LCM [1] :  Plans in a sentence-embedding (concept) space utilizing SONAR embeddings and decodes to text afterward, showing strong results on open-ended generation (e.g., summarization/expansion) and multilingual transfer. Unlike our post-training adaptor, LCM is a pretrained concept-space LM.
> >
> > - CoCoMix [4] : Pretrains with continuous “concepts” (from a sparse autoencoder) interleaved with token states to improve efficiency/interpretability. This is similarly a pretraining-time approach.
> >
> > Also, the most recent relevant one that we found is:
> >
> > - CALM [5]  (Shao, Oct, 2025): It replaces next-token with next-vector prediction, using a high-fidelity autoencoder to compress K tokens → 1 continuous vector, which is similar to our Sem variant from using semantic embeddings. Yet this pre-trains model from these embeddings acquired from autoencoder.
> >
> >
> > In fact, within reasoning compression, many recent efforts pretrain models directly in such embedding spaces; because it’s easier than a model with less plasticity to adapt to a new paradigm, as evidenced by the challenge in Coconut to extending to Llama-3 family models.
> >
> > Indeed, our path (i.e. post-training) can be harder to adapt; yet we show early signs of lifting the abstraction level of a pretrained model’s building blocks could be done post-training, which we believe opens exciting research avenues.
> >
> > ---
> >
> > Finally, we sincerely thank the reviewers for their time, effort, and constructive feedback.

---

### Official Review · Reviewer_yGxr · 2025-11-01

**Soundness:** 3
**Presentation:** 3
**Contribution:** 3
**Rating:** 4
**Confidence:** 3

**Summary:**

This study addresses the issues of inefficient computation and suboptimal reasoning granularity encountered by LLMs when generating step-by-step reasoning chains on a token-by-token basis. To this end, the authors introduce a novel framework designed to lift the abstract reasoning capabilities of pre-trained language models from the token level to the sentence level. The central concept of this framework is to have the model autoregressively predict the continuous embedding vectors of the next sentence, instead of generating raw tokens. Specifically, the work develops two types of embedding vectors—semantic embeddings and contextual embeddings—and investigates both discretized and continuous inference modes. Experimental findings demonstrate that, under the setting of “contextual embeddings with continuous inference”, the proposed method achieves performance comparable to CoT, while simultaneously reducing the inference computational cost by half.

**Strengths:**

1. The paper is clearly written, and the figures are well-designed and easy to follow.
2. The attempt to perform auto-regressive decoding at the sentence level is innovative.

**Weaknesses:**

1. I understand the authors' experimental scope was limited to GPT-2-sized models due to computational constraints. To strengthen the experimental evidence, I suggest the authors broaden their evaluation to include more recent models, such as Llama-3.2-1B-Instruct, Qwen3-0.6B, and Qwen3-1.7B.
2. I am puzzled by the mere 2x speedup achieved for sentence-level auto-regressive decoding. Given the degree of parallelism involved, a more significant gain (e.g., 10x) would be expected intuitively. I would appreciate a detailed explanation or analysis from the authors to clarify why the speedup was only twofold.
3. Could the authors clarify what the bolded figures in Table 2 specifically signify? My interpretation is that they are intended to highlight the best-performing method within each dataset. However, based on the data presented, it appears the best method has been incorrectly marked across all four datasets. The authors should verify and correct these annotations to ensure the table's accuracy.

**Questions:**

Please see the weakness.

---

> ### Author Response · Authors · 2025-12-01
>
> Dear Reviewer, thank you for your detailed review.
>
> While the review system does not allow us for further communication, below we address your concerns to our best effort.
>
> ---
>
>
> **P1. I understand the authors' experimental scope was limited to GPT-2-sized models due to computational constraints. To strengthen the experimental evidence, I suggest the authors broaden their evaluation to include more recent models, such as Llama-3.2-1B-Instruct, Qwen3-0.6B, and Qwen3-1.7B.**
>
> Indeed, after the submission, we have tried our framework with Qwen-3-0.6B and Llama-3.2-1B, yet we had difficulty of replicating the results with our framework that could match the performance of CoT. While we saw initial promising signs with stronger adaptation priors (i.e. KL Divergence instead of CE loss), we leave this to the future work. Nevertheless, as we highlight, this paper is a first step, probing whether models can form and exploit their own latent embeddings for higher-level reasoning, and we believe that with additional regularization smoothly adapting extensively pre-trained models to the latent mode could be feasible.**
>
> ---
>
> **P2. I am puzzled by the mere 2x speedup achieved for sentence-level auto-regressive decoding. Given the degree of parallelism involved, a more significant gain (e.g., 10x) would be expected intuitively. I would appreciate a detailed explanation or analysis from the authors to clarify why the speedup was only twofold.**
>
> As noted in Appendix C, the Pure Latent decoding mode theoretically offers massive reductions in both Attention complexity and MLP overhead compared to CoT:
>
> *Attention Complexity:*
>
> - CoT: $\mathcal{O}(L^2 R^2)$ (Quadratic w.r.t total tokens)
> - Pure Latent: $\mathcal{O}(R^2)$ (Quadratic w.r.t reasoning steps only)
>
> *MLP Overhead:*
>
> - CoT: $\mathcal{O}(LR)$ (Linear w.r.t total tokens)
> - Pure Latent: $\mathcal{O}(R)$ (Linear w.r.t reasoning steps only)
>
> However, the simplified complexity analysis above omits a crucial constant common to all methods: the overhead incurred by the input prompt. In our experiments, the prompt length $N_0$ is significant. This introduces a large fixed cost $\mathcal{O}(C)$, encompassing both the prefill compute and the attention to the prompt during decoding. This large constant dominates the total runtime in our current regime, diluting the relative speedup from the theoretical order-of-magnitude gain to the observed $\approx 2\times$.
>
> We posit that as the generation length increases (i.e., with longer CoT), the fixed prompt cost will become less dominant, and the empirical speedup will asymptotically approach the theoretical gap (e.g., the $10\times$ gain as described).
>
> ---
>
> **P3. Could the authors clarify what the bolded figures in Table 2 specifically signify? My interpretation is that they are intended to highlight the best-performing method within each dataset. However, based on the data presented, it appears the best method has been incorrectly marked across all four datasets. The authors should verify and correct these annotations to ensure the table's accuracy.**
>
> Thank you for the feedback. We have revised Table 2 for better readability. Bolded numbers mark the best-performing method **among our proposed variants** within each dataset. To avoid confusion, external baselines (CoT, CoCoNut) are highlighted with a distinct background color. We have also updated the caption to explicitly state these conventions.
>
> ---
>
> Finally, we really appreciate your time and effort in reviewing our paper..

---

### Official Review · Reviewer_v7p4 · 2025-11-01

[review text omitted: it was posted to a different submission]

---

> ### Author Response · Authors · 2025-11-13
> **Wrong Review**
>
> Dear reviewer,
>
> We believe that an incorrect review may have been submitted in error.
> If this was unintentional, we would greatly appreciate if you could kindly revise and update your review accordingly.
> Your review would be of great help to us.
>
> Thank you for your time and understanding.
>
> sincerely,
> authors

---

> > ### Comment · Reviewer_v7p4 · 2025-11-15
> >
> > i think the system has a glitch, plz check again

---

> ### Author Response · Authors · 2025-12-01
>
> Dear Reviewer,
> Thank you for your time. Now the issue has been resolved, and below we address your concerns to our best effort.
>
> ---
>
> **P1. The paper acknowledges but doesn't fully address the fragility of continuous embeddings to perturbations, which could limit practical deployment.**
>
> The fragility of continuous embeddings is largely mitigated in our setup by a language-level bottleneck: we decode last hidden states to text and then re-encode into the latent space, which limits error accumulation across steps. This problem is largely also reported in previous works like [1] [2] That said, our method yields some possible directions, e.g., exploring mild discretization/quantization of the latent space, noise-aware/denoising objectives, or lightweight error-correction mechanisms.
>
>
> ---
>
>  **P2. How does performance scale with larger pretrained models (e.g., 7B+ parameters)?**
>
> We hypothesize that scaling hinges less on parameter count and more on plasticity vs. pretraining depth. At GPT-2 scale, size itself did not appear decisive; rather, models with stronger token-space priors are harder to adapt to latent reasoning.
>
> On our post-submission experiments with Qwen-3-0.6B and Llama-3.2-1B, we had difficulty of replicating the results with our framework that could match the performance of CoT. We saw initial promising signs with stronger adaptation priors (*i.e.* KL Divergence instead of CE loss) but we leave this to the future work. For now, we explore a general direction towards how abstraction over the existing building blocks of autoregressive models could be framed.
>
> ---
>
> *[1] The pitfalls of imitation learning when actions are continuous (Simchowitz, 2025)}*
>
> *[2] Large concept models (LCM team, 2025)*

---

### Official Review · Reviewer_7gE9 · 2025-11-03

**Soundness:** 2
**Presentation:** 3
**Contribution:** 3
**Rating:** 4
**Confidence:** 4

**Summary:**

This paper proposes a new paradigm for language model reasoning, called Sentence-level Latent Reasoning. Unlike traditional token-level autoregressive generation, this approach enables the model to perform prediction and reasoning in the sentence embedding space, thereby achieving a more abstract and efficient reasoning process.

**Strengths:**

1. The paper introduces a novel idea of performing autoregressive reasoning in the sentence embedding space, elevating the reasoning hierarchy of language models from the token level to the sentence level.
2. The proposed SentenceLens enables decoding intermediate latent states into natural language sentences, making the model’s “thought process” interpretable and analyzable.
3. By reasoning directly in the continuous embedding space, computational efficiency is significantly improved (1.5–2.5× speedup).

**Weaknesses:**

1. Although the paper proposes the concept of “sentence-level reasoning,” it does not sufficiently justify why sentence-level embeddings can necessarily capture the logical structures required for reasoning that token-level embeddings cannot.
2. The comparison is quite limited: it only benchmarks against one latent reasoning model (Coconut), and shows no significant advantage except on the Blocksworld task (where the performance gap is unusually large, raising concerns about possible evaluation bias). Additionally, computational efficiency is not compared with Coconut.

**Questions:**

1. Does your method offer any computational efficiency advantage compared to Coconut? If so, by how much?
2. Why do you introduce an encoder during training? Why are its parameters shared with the decoder?

---

> ### Author Response · Authors · 2025-12-01
>
> Dear Reviewer, thank you for your detailed review. While the review system does not allow us for further communication, below we address your concerns to our best effort.
>
> ---
>
> **P1. Although the paper proposes the concept of “sentence-level reasoning,” it does not sufficiently justify why sentence-level embeddings can necessarily capture the logical structures required for reasoning that token-level embeddings cannot.**
>
> Our claim is not that tokens cannot encode logic, but that reasoning operations (e.g., introduce a premise, apply a rule, resolve a variable, conclude) often are naturally sentence-scale transitions. Constraining the model to predict the next sentence incentivizes it to learn stepwise invariants (entailment, consistency, variable binding) that are hard to diffuse at the token scale.
>
> Note that in ProsQA, all our model variants still yield better results than CoT, despite they do not necessarily encode and carry on multiple states as the benefit implied by the Coconut when using language-level decoding (i.e. decoding back to language and then re-encoding). Therefore, from here, we can deduce the effect that sentence-level embeddings often capture the logical structures required for reasoning that token-level embeddings cannot.
>
> ---
>
> **P2. The comparison is quite limited: it only benchmarks against one latent reasoning model (Coconut), and shows no significant advantage except on the Blocksworld task (where the performance gap is unusually large, raising concerns about possible evaluation bias). Additionally, computational efficiency is not compared with Coconut.**
>
> **Does your method offer any computational efficiency advantage compared to Coconut? If so, by how much?**
>
> Our work was largely motivated by Coconut and can be seen as an extension of it; we intentionally use GPT-2 as in their setup while addressing limitations in their framework. We generally regard CoT as an upper bound in many domains (except Logical), so the core problem is how to adapt pretrained models into latent-reasoning models without losing prior such that it would match the performance of CoT. Coconut tackles this with an N-stage curriculum that requires N forward passes, but this design is hard to scale and falls much behind especially in GSM8K.
>
> By contrast, our method offers significant computational efficiency in both training and testing: we need one forward pass during training, and at inference we flexibly adjust the latent budget to match problem difficulty, whereas Coconut uses a fixed, deterministic budget of N latent states and must always forward N times. Performance. Our method yields better results than Coconut on all but ProsQA (Logic); in the Logical domain, prior methods like iCoT reach ~99%, suggesting that gradual transition from token to latent is essential, yet our approach still achieves very strong results vs CoT.
>
> ---
>
> **P3. Why do you introduce an encoder during training? Why are its parameters shared with the decoder?**
>
> We introduce an encoder–decoder to obtain a latent embedding that serves as the input/target for our latent reasoning model. The reviewer is right that, in principle, we could feed the decoder’s last-token hidden state (as in Coconut) instead of an autoencoder-derived embedding. We chose the autoencoder route because relying on last-token states couples reasoning to token-level decoding and hinders parallelizable training; Coconut must obtain those states sequentially.
>
> In contrast, our encoder provides a compact, pre-computable sentence embedding that we can prepare ahead of time and reuse, which makes training more parallel and efficient. We also expect a well-formed latent space to better encapsulate sentence structure, providing a clearer target for latent supervision.
>
> Weight sharing is optional. We tie the encoder and decoder weights to reduce memory usage during training and to keep the latent space aligned with the decoder’s distribution. A non-shared variant is feasible and behaves similarly, but with higher memory cost.
>
> ---
>
> Finally, we really appreciate your time and effort in reviewing our paper.

---

### Official Review · Reviewer_rMkZ · 2025-11-03

**Soundness:** 2
**Presentation:** 3
**Contribution:** 2
**Rating:** 4
**Confidence:** 4

**Summary:**

This paper proposes a framework for performing latent reasoning in the sentence embedding space. The authors repurpose a pretrained language model (GPT-2) and construct two types of sentence-level embeddings (semantic and contextual) to map natural language into a continuous latent space, enabling autoregressive generation within that space. The paper further introduces a visualization tool, *SentenceLens*, to analyze the model’s “latent reasoning trajectories” in embedding space.

**Strengths:**

- The experiments span diverse reasoning domains (mathematical, logical, commonsense, and planning), demonstrating the general applicability of the framework.

- The paper provides an interpretability tool (*SentenceLens*) for analyzing latent reasoning trajectories.

- Despite the complex design, the authors’ exposition is overall clear and logically structured.

**Weaknesses:**

- The InfoNCE loss ratio and λ parameter for CTX-C are not specified, making the experiments hard to reproduce.

- The scope of fine-tuning (which components are updated) is unclear, and gradient flow is not described.

- No comparisons are made against strong latent-reasoning baselines such as CoCoMix, CoDi, or Token Assorted.

- Table 2 reports only a single inference mode; it should separately present accuracy and FLOPs across different reasoning modes.

- The empirical improvements over coconut and cot are relatively minor and do not convincingly demonstrate the claimed efficiency–performance trade-off.

**Questions:**

- Is $h_N$ the final-layer hidden state or a multi-layer aggregation?

- What value of λ is used for CTX-C? Have you performed a λ-sweep analysis?

- How does *SentenceLens* decode embeddings originating from different encoders?

- What is the exact architecture of the M-projection head? Is it used during both training and inference?
 How does the Continuous mode avoid semantic drift without the M head?

- Which components are frozen during fine-tuning? Do the encoder and decoder receive gradient updates?

- Why are CoCoMix, CoDi, and Token Assorted excluded from the baseline comparisons?

---

> ### Author Response · Authors · 2025-12-01
>
> Dear Reviewer, thank you for your detailed review. While the review system does not allow us for further communication, below  we address your concerns to our best effort.
>
> ---
> **P1: The empirical improvements over coconut and cot are relatively minor and do not convincingly demonstrate the claimed efficiency–performance trade-off.**
>
> Our paper’s main purpose was to provide a preliminary study on whether latent reasoning with sentence embedding could match the performance of a model fine-tuned to output CoT reasoning token by token. Our method provides early signs of pretrained models possibly could adapt to such continuous space using sentence embeddings. Therefore, we do not mainly aim to perform better than CoT, but rather reach a similar performance to the level of CoT while bringing compute efficiency, which would be much easier for the model to learn as it matches the NTP objective of pretraining.
>
> For Coconut, we observe gains in all other domains except for the logical domain, and most importantly, we bring two significant benefits over Coconut. First, training only requires just one forward pass whereas Coconut requires for N forward passes which significantly hinders training speed and limits scalability. Second, we also provide much more interpretability as our latent hidden states at each position is directly decodable using the decoder whereas Coconut only has to indirectly hypothesize using logit lens.
>
> ---
>
> **P2: The InfoNCE loss ratio and λ parameter for CTX-C are not specified, making the experiments hard to reproduce.
> What value of λ is used for CTX-C? Have you performed a λ-sweep analysis?**
>
> For reproduction, we have provided a **full anonymized github repository link** in our original submission.
>
> For CTX-C, we have used a weight of 0.5, based on our early results when experimenting with lambda = [0.1, 0.5, 1.0]. We believe lambda-sweep analysis could have been indeed beneficial, but we thought that compared to other parts of the paper, the lambda sweep analysis could be minor details.
>
> for InfoNCE loss ratio, we used 1.0, based on our early observation that within reasonable range i.e. [0.1~1.0], the two losses, i.e. CE loss and InfoNCE loss fairly decrease together.
>
> ---
>
> **P3: The scope of fine-tuning (which components are updated) is unclear, and gradient flow is not described.**
>
> For section 2, all encoder / decoder ( which share weight, and thus is just one model) are updated.
>
> For section 3, for training our latent model which reasons sentence-by-sentence, the encoder / decoder (which again is a single model) is frozen, and only the latent model is updated.

---

> > ### Author Response · Authors · 2025-12-01
> >
> > **P4. No comparisons are made against strong latent-reasoning baselines such as CoCoMix, CoDi, or Token Assorted.
> > Why are CoCoMix, CoDi, and Token Assorted excluded from the baseline comparisons?**
> >
> > This is a fair point that we have only compared against Coconut. However, all these works have not yet open-sourced their code [CocoMix, Codi, Token Assorted] which makes the comparison difficult. Also CocoMix pretrains the model from the scratch and thus is hard to compare apple-to-apple to our method.
> >
> > For others, we cannot provide a comparison in performance; instead, we provide explanation as to how it differs from our method.
> >
> > > **Codi**: Uses KL-style distillation at the final position from a teacher and runs a fixed number of latent passes; ours is better because it adapts the number of steps to the task and handles longer chains without a fixed budget.
> >
> > > **Token Assorted**: Replaces early CoT with VQ-VAE discrete tokens to shorten traces, but at the long term, ours is more sustainable and scalable as fixed-size codebook will be nonetheless limited by its expressivity.
> >
> > ---
> >
> > **P5. Is $h_{N}$ the final-layer hidden state or a multi-layer aggregation?**
> >
> > $h_{N}$ is the final layer hidden state.
> >
> > ---
> >
> > **P6. What is the exact architecture of the M-projection head? Is it used during both training and inference? How does the Continuous mode avoid semantic drift without the M head?**
> >
> > We believe that you are referring to this part. “shallow projection layers between the encoder output and latent model input, and between the latent model output and decoder input.”
> >
> > This is a 2-layer MLP with GeLU activation function in between. This is used during both training and inference.
> >
> > Continuous mode avoid semantic drift without the projection layers as in that case, because the infoNCE loss encourages the *unprojected* representations to lie in the same semantic space. The projection layers exist to better facilitate mapping
> >
> > - encoder output space -> latent model input space
> > - Latent model output space -> decoder input space.
> >
> > ---
> >
> > **P7. Table 2 reports only a single inference mode; it should separately present accuracy and FLOPs across different reasoning modes.**
> >
> > Please note that In table 2, we are reporting both inference modes, i.e. continuous and discretized.
> >
> > Also, we are reporting FLOPS of continuous mode only in Table 3, because as we highlight, discretized mode (i.e. which explicitly decodes into language and encodes again) incurs extra computation overhead and brings computation gain when the reasoning chain is sufficiently long ( as in blocksworld), which we highlight as “, we highlight that even DISCRETIZED inference outperform CoT in longer reasoning tasks (e.g., Blocksworld w/ average trace length R ∼ 9.1: 52.26 GFLOPs vs. 58.69 GFLOPs). We expect this efficiency gap to grow as the length of reasoning trace increases.”  Therefore we have only reported the inference mode for continuous cases.
> >
> > ---
> >
> > Finally, we really appreciate your time in reviewing our manuscript.

---

### Meta-Review · Area_Chair_wzRA · 2026-01-07

**Summary:**

Major concerns include:
1. Lack of clarity in paper writing: multiple technical and experimental details that are considered critical by reviewers are missing. Reviewers are thus concerned about the reproducibility of this work and are unable to judge the value of the proposed approach. Reviewers also feel unclear about the intuition of the proposed work.
2. Stronger baselines, particularly those about latent reasoning, are needed.
3. Less convincing experimental results, given the marginal performance improvement of the proposed work over existing baselines and the less interpretable efficiency gain.
4. Need for experimenting with larger language models than GPT-2.

**Reviewer Concerns:**

Most concerns were addressed (including those not listed above), but,
- Regarding the baselines (e.g., CODI and CoCoMix), the authors claimed that the source code of these baselines is unavailable. However, as I checked, these baselines have source code on GitHub from early or mid 2025.
- Regarding the result interpretation, the authors claimed that their goal is not to have their approach outperform the baselines. Instead, they aim to showcase the potential of latent reasoning via sentence embeddings. However, how significant this contribution is could be subject to each reviewer's judgement.
- Regarding experimenting with larger LMs, the authors did not provide new results due to an implementation issue they encountered.

**Reviewer Scores:**

It is a little hard to tell if reviewers will increase their score, given that several critical concerns are not addressed (although the authors did address many others well).

---

### Decision · Program_Chairs · 2026-01-26

Reject